# Modelling conformational state dynamics and its role on infection for SARS-CoV-2 Spike protein variants

**Natália Teruel**[1], **Olivier Mailhot**[1,2], **Rafael J. Najmanovich**[1] *

**1** Department of Pharmacology and Physiology, Faculty of Medicine, Université de Montréal, Montreal, Canada, **2** Institute for Research in Immunology and Cancer (IRIC), Faculty of Medicine, Université de Montréal, Montreal, Canada

* rafael.najmanovich@umontreal.ca

## Abstract

The SARS-CoV-2 Spike protein needs to be in an open-state conformation to interact with ACE2 to initiate viral entry. We utilise coarse-grained normal mode analysis to model the dynamics of Spike and calculate transition probabilities between states for 17081 variants including experimentally observed variants. Our results correctly model an increase in open-state occupancy for the more infectious D614G via an increase in flexibility of the closed-state and decrease of flexibility of the open-state. We predict the same effect for several mutations on glycine residues (404, 416, 504, 252) as well as residues K417, D467 and N501, including the N501Y mutation recently observed within the B.1.1.7, 501.V2 and P1 strains. This is, to our knowledge, the first use of normal mode analysis to model conformational state transitions and the effect of mutations on such transitions. The specific mutations of Spike identified here may guide future studies to increase our understanding of SARS-CoV-2 infection mechanisms and guide public health in their surveillance efforts.

## Author summary

The present work explores the molecular mechanisms underlying and potentially helping new strains of SARS-CoV-2 to gain an evolutionary advantage during the ongoing COVID-19 pandemics. We show how a computational method called normal mode analysis that treats protein dynamics in a simplified manner is capable to predict the higher propensity of the Spike protein to be in the open state in which it is capable to interact with the human ACE2 receptor and thus facilitate cell entry. Because the simulation of the simplified computational model is relatively less demanding on resources than alternative methods, we were able to simulate over 17000 mutations in the SARS-CoV-2 Spike protein to identify multiple mutations that if they were to appear as the virus continues to evolve, could confer an evolutionary advantage. As a matter of fact, our predictions foresaw the emergence of particular mutations such as N501Y that appeared in several variants of concern. Our results can inform public health regarding new variants and serves as a proof of concept for the application of normal mode analysis to study the effect of

**Data Availability Statement:** Raw data and structures used to build the images presented here are available in a Github repository (https://github.com/nataliateruel/data_Spike). All vibrational

entropy results are available for visualisation and analysis through a link to the dms-view open-access tool, available on GitHub through the same URL above. The Najmanovich Research Group Toolkit for Elastic Networks (NRGTEN) including the latest ENCoM implementation is freely available at https://github.com/gregorpatof/nrgten_package.

**Funding:** OM is the recipient of a PhD fellowship from the Fonds de Recherche du Québec - Nature et Technologie (FRQ-NT). This work was funded by grants from Genome Canada (http://genomecanada.ca), Genome Québec (https://www.genomequebec.com) as well as the Natural Sciences and Engineering Research Council (NSERC) grant number RGPIN05332-2019. This research was enabled in part by support provided by (Calcul Québec) (https://www.calculquebec.ca/) and Compute Canada (www.computecanada.ca). The funders had no role in study design, data collection and analysis, decision to publish, or preparation of the manuscript.

**Competing interests:** The authors have declared that no competing interests exist.

mutations on both, protein dynamics and conformational transitions in a high-through-put manner.

## Introduction

The coronavirus pandemic has emerged as a major and urgent issue affecting individuals, families and society as a whole. Among all outbreaks of aerosol transmissible diseases in the 21st century, the COVID-19 pandemic, caused by the severe acute respiratory syndrome coronavirus 2 (SARS-CoV-2) virus [1,2], has the highest infection and death cumulative numbers—61 million infections and over 1.4 million deaths, according to the World Health Organization (WHO) epidemiological report of December 1, 2020 [3]. Recent WHO reports also show significant weekly increases in the number of infections and deaths as countries start to face upcoming waves of the disease. In 2003 the SARS coronavirus (SARS-CoV) pandemic caused 8,098 infections and 774 deaths before it was brought under control [4,5]. In 2012, the Middle East respiratory syndrome-related coronavirus (MERS-CoV) outbreak caused 2499 infections and 858 deaths, presenting the highest fatality rate [6]. SARS-CoV-2, SARS-CoV and MERS-CoV, as coronaviruses in general, present considerable mutation rates, which may contribute to future outbreaks. For instance, SARS-CoV-2 is estimated to have a mutation rate close to the ones presented by MERS-CoV [7] and by SARS-CoV [8], as well as other RNA viruses, showing a median of $1.12 \times 10^{-3}$ mutations per site per year [9]. The high mutation rate may in part be responsible for the zoonotic nature of these viruses and points to a clear risk of still-undetected additional members of the *coronavirideae* family of viruses making the jump from their traditional hosts to humans in the future.

The SARS-CoV-2 Spike protein (Uniprot ID P0DTC2) is responsible for anchoring the virus to the host cell. The entry receptor for SARS-CoV-2 and other lineages of human coronaviruses is the human cell-surface protein angiotensin converting enzyme 2 (ACE2) (Uniprot ID Q9BYF1) [10]. Therefore, studying the Spike protein family is essential to understand the evolution of coronaviruses.

SARS-CoV-2 Spike is a homo-trimeric glycoprotein, with each chain built by subunits S1 and S2, delimited by a Furin cleavage site at residues 682–685. The S1 subunit comprises the N-terminal domain (NTD), located in the peripheral part of the extramembrane extreme, and the receptor-binding domain (RBD), the most flexible site, located in the central part of this same extreme. The S2 subunit consists of the fusion peptide (FP), heptad repeat 1 (HR1), heptad repeat 2 (HR2), the transmembrane domain (TM), and the cytoplasmic tail (CT) (Fig 1). The interaction between Spike and ACE2 relies on Spike being in its open conformation, in which the receptor-binding domain (RBD) is extended [11]. The study of the binding properties between Spike and ACE2, although important, cannot explain all the nuances of the infection mechanism. An example of this limitation is the comparison between SARS-CoV and SARS-CoV-2, which have different rates of infection even though they share similar Spike-ACE2 affinities [12]. These facts lead us to consider the contribution of Spike protein dynamics to the infection process.

Computational structural biology methods have grown in both accuracy and usability over the years and are increasingly accepted as part of an integrated approach to tackle problems in molecular biology. Such integration speeds up research, decreases needs in infrastructure, reagents, and human resources and allows us to evaluate increasingly larger data sets. Computational approaches are being extensively used in the study of SARS-CoV-2 and its mechanisms of infection [13–15]. Among these, we highlight the study of dynamic properties of the

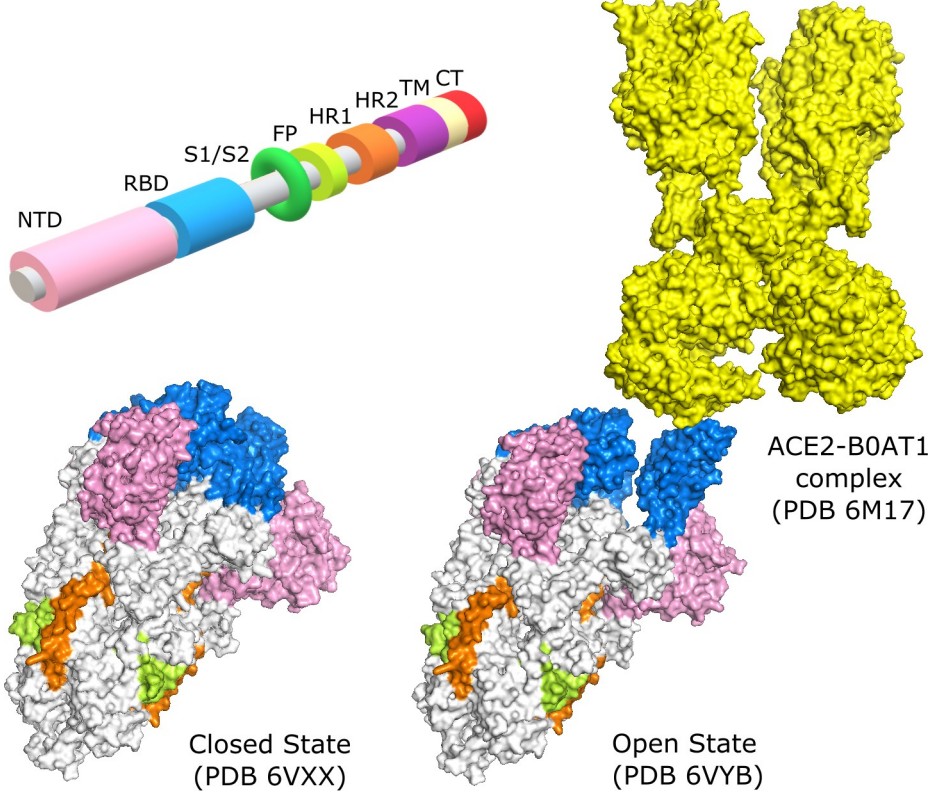

**Fig 1. Domains of the Spike protein.** N-Terminal Domain (NTD), Receptor Binding Domain (RBD), Subunit 1/ Subunit 2 junction (S1/S2), Fusion Peptide (FP), Heptad Repeat 1 (HR1), Heptad Repeat 2 (HR2), Transmembrane Domain (TM), and the Cytoplasmic Tail (CT). Crystallography structure in the conformational state of all 3 RBD domains closed (PDB 6VXX) and of 1 RBD open (PDB 6VYB), binding to ACE2 (PDB 6M17).

Spike protein as well as in antibody recognition and the search for therapeutic interventions [16–18].

Several aspects of Spike protein dynamics are currently being studied, with a range of particular goals: to evaluate the docking of small molecules to the RBD domain [19], to search for alternative target binding-sites for vaccine development [20], to understand residue-residue interactions and their effects on conformational plasticity [21] and to investigate the flexibility of different domains in particular conformational states [22].

Various combinations of normal mode analysis (NMA) and molecular dynamics (MD) methods are being employed in the study of different conformational states [23] and of different coronavirus variants [24]. These methods, however, are limited with respect to their ability to study the effects of mutations on dynamics since they are either extremely taxing on computational resources in the case of MD or agnostic to the nature of amino acids in the case of traditional coarse-grained NMA. In the past, our group developed a coarse-grained NMA model called ENCoM (for Elastic Network Contact Model) that considers the chemical nature of amino acids and their interactions and consequently their effect on dynamics [25]. This makes ENCoM perform better than other NMA models on traditional applications but more importantly makes it the only coarse-grained NMA model capable of predicting the effect of mutations on protein stability and function as a result of dynamical properties [26–28]. As a coarse-grained NMA model, ENCoM is not much costlier to run than other traditional NMA models and this makes it very attractive as a tool to screen the effect of mutations in a high-throughput manner.

In the present study, we use ENCoM to study the dynamics of the Spike protein, considering different conformational states and several sequence variants observed during the current pandemic, as well as through large-scale analysis of *in silico* mutations. Experimental analysis of the effect of the SARS-CoV-2 Spike mutation D614G and the comparison between SARS-CoV and SARS-CoV-2 Spike proteins show unique dynamic characteristics that correlate with epidemiological and experimental data on infection. The present work shows that we can replicate such results computationally, suggesting that rigidity or flexibility of different Spike conformational states affects infectivity. We present a high throuhput analysis of simulated single amino acid mutations on dynamical properties to seek potential hotspots and individual Spike variants that may be more infectious and therefore may guide public health decisions if such variants were to appear in the population. We also introduce a Markov model of occupancy of molecular states with transition probabilities derived from our analysis of dynamics that recapitulates experimental data on conformational state occupancies. This is the first application of an NMA method that derives transition probabilities from normal modes and employs them in a dynamic system to predict the occupancy of different conformational states. We model the occupancy of several variants and highlight those that may be useful in studying future epidemiological trends that could be responsible for new outbreaks and lastly, we expand and apply the methodology to multiple mutations in cases of variants of concern that that were observed during the COVID-19 pandemic with their full complement of Spike mutations.

## Materials and methods

### Spike protein models

We performed our analyses using the crystallographic models of the SARS-CoV-2 Spike protein in the open (PDB ID 6VYB) and closed (PDB ID 6VXX) states. The open (prefusion) state was designed with an abrogated Furin S1/S2 cleavage site and two consecutive proline mutations that improve expression [29]. Despite the mutations, the engineered structures correctly represent the conformational states of Spike, as confirmed by independently solved structures [30]. The PDB structures used for the SARS-CoV comparison were 5X58 and 5X5B for closed state and one RBD open state, respectively [31].

We removed heteroatoms, water molecules, and hydrogen atoms from the PDB structures. Missing residues were reconstructed using template-based loop reconstruction and refinement with Modeller [32]. Single amino acid mutants were generated using FoldX4 [33]. Vibrational Difference Score (VDS, defined below) and occupancy calculations were performed with reconstructed closed and one-RBD-open structures using as template 6VXX and 6VYB. These engineered structures contain the GSAS sequence in the Furin cleavage site as well as two prolines in positions 986 and 987. In order to minimize potential artefacts in the calculations due to modelling errors, we chose to model all mutations and perform subsequent calculations using the above engineered structures and sequences unless otherwise noted. That is to say, when we refer to the wild type SARS-CoV-2 Spike protein in our calculations, it is the Spike protein with the above alterations in the Furin clivage site as well as the pair of prolines. This choice in our methodology is made as stated to decrease the possibility of modelling artefacts as the alternative would have required modelling 6 additional mutations to 'de-engineer' the structures of the open and closed states.

For the parameter fitting used in the calculation of occupancies, we utilized the following experimentally determined structures for which occupancy data exists as follows (acronyms described in results): S-GSAS/WT: 7KDG,7KDH; S-GSAS/D614G: 7KDI,7KDJ [30]; S-R/x2: 6ZOX; S-R/PP/x1: 6ZOY,6ZOZ; S-R: 6ZP0; S-R/PP: 6ZP1,6ZP2 [34].

## Dynamic analyses

We analysed dynamic properties of the Spike protein with ENCoM [25]. ENCoM employs a potential energy function that includes a pairwise atom-type non-bonded interaction term and thus makes it possible to consider the effect of the specific nature of amino-acids on dynamics. Normal mode analysis (NMA) explores protein vibrations around an equilibrium conformation by calculating the eigenvectors and eigenvalues associated with different normal modes [35–37]. Representing each protein residue as a single point, for a given conformation of a protein with N amino acids, we obtain 3N - 6 nontrivial eigenvectors. Each eigenvector represents a linear, harmonic motion of the entire protein in which each amino acid moves along a unique 3-dimensional Euclidean vector. The associated eigenvalues rank the eigenvectors in terms of energetic accessibility, lower values corresponding to global, more easily accessible motions.

NMA calculations allow us to computationally estimate b-factors associated with the protein structure, as shown in Eq 1 for the $i^{th}$ residue, which in turn are related to local flexibility. Higher predicted b-factors denote more flexible positions. Individually calculated b-factors are combined in a vector for a protein sequence or part thereof and called Dynamical Signature.

$$B_i = \sum_{n=7}^{3N} \frac{E_{n,i,x}^2 + E_{n,i,y}^2 + E_{n,i,z}^2}{\lambda_n} \tag{1}$$

The eigenvectors and associated eigenvalues can also be used to obtain the vibrational contribution of the entropic components of the free energy. Vibrational entropy [38] is calculated as described in Eq 2 in units of $J.K^{-1}$, where N is the total number of amino acids in the protein, $v_i$ is the vibrational frequency and $K_B$ is the Boltzmann constant. Eq 3 shows the association between eigenvalues and vibrational frequency.

$$S_{vib} = K_B \sum_{n=7}^{3N} \left\{ \frac{\beta v_n}{e^{\beta v_n} - 1} - \ln\left[1 - e^{-\beta v_n}\right] \right\} \tag{2}$$

$$\lambda_n = v_n^2 \tag{3}$$

Measuring the difference of vibrational entropy ($\Delta S_{vib}$) between a mutant and a wild type (WT), one can calculate how much a mutation affects the overall flexibility and stability of the mutant relative to the WT. The $\Delta S_{vib}$ value predicted by ENCoM is negative when the mutation makes the protein more flexible and positive when the mutation makes the protein more rigid. Vibrational entropy calculations are dependent on the thermodynamic β factor, that for pseudo-physical models such as ENCoM serves as a scaling factor. This term was optimized to fit experimental Gibbs free energy differences [39] and established as β = 1. The differences between the $\Delta S_{vib}$ values for closed and open states, which we call Vibrational Difference Score (VDS), were calculated for each mutant (VDS = $\Delta S_{vib (open)} - \Delta S_{vib (closed)}$) as a means to select mutations of interest. A positive VDS suggests that the mutation makes the open state less flexible and/or the closed state more flexible, favouring the open conformation relative to the WT. Conversely, a negative VDS suggests that the mutation favours the closed state more so than the WT. The computational cost of obtaining the VDS (modelling the mutation with FoldX on both states and computing $\Delta S_{vib}$ values for each with ENCoM) is approximately 20 CPU-minutes per mutant, making the total cost of these computations for the 17 081 mutants considered in this work around 0.65 CPU-year. Of the 20 CPU-minutes per mutant, around 2 minutes are spent on running FoldX and the rest on running ENCoM.

The Najmanovich Research Group Toolkit for Elastic Networks (NRGTEN) [39], with the latest implementation of ENCoM, also includes a function to evaluate state occupancies by

calculating transition probabilities between different states. A probability $P_j$ of moving along each eigenvector $j$ can be obtained using a Boltzmann distribution given its associated eigenvalue $\lambda_j$ and a scaling factor $\gamma$.

$$P_j = \frac{e^{-\lambda_j/\gamma}}{\sum_{i=7}^{3N} e^{-\lambda_i/\gamma}} \tag{4}$$

Let us consider two conformations A and B of the same protein and the vector $\mathbf{E}_{A\rightarrow B}$, which represents the conformational change going from conformation A to conformation B. The overlap between each normal mode $\boldsymbol{M}_j$ computed from conformation A and the $\mathbf{E}_{A\rightarrow B}$ vector is a value between 0 and 1 describing how well that normal mode recapitulates the conformational change required to go from one state to the other [40].

$$O\left(\boldsymbol{E}_{A\rightarrow B}, M_j\right) = \frac{\left|\boldsymbol{E}_{A\rightarrow B} \cdot \boldsymbol{M}_j\right|}{\|\boldsymbol{E}_{A\rightarrow B}\|\|\boldsymbol{M}_j\|} \tag{5}$$

We can then calculate the transition probability of going from conformation A to conformation B as the weighted sum of the Boltzmann probability $P_j$ of each normal mode $\boldsymbol{M}_j$ times the overlap between that normal mode and the conformational change $E_{A\rightarrow B}$.

$$P_{A\rightarrow B} = \sum_{j=7}^{3N} P_j \times O\left(\boldsymbol{E}_{A\rightarrow B}, \boldsymbol{M}_j\right) \tag{6}$$

The reverse probability $P_{B\rightarrow A}$ can be computed in the same fashion, giving an indication of which conformation is favored between the two.

A simple way of computing the occupancies of these conformations from the transition probabilities is to use a Markov model. Each conformation is represented by a state, and the transition probabilities between states are computed as described above. We add a constant $k$ to all states as the probability of staying in that state. Since all states must have outgoing transition probabilities that sum to 1, we normalize these values after the addition of $k$. For a two-state Markov chain representing the open and closed states of the Spike protein, we obtain the diagram shown in Fig 2. All transition probabilities are computed using ENCoM and Eq 6. The parameters $k$ and $\gamma$ need to be optimized for the system being studied as they are not directly coupled to physical quantities because of the pseudo-physical, coarse-grained nature of the ENCoM model. Once the parameters are set, there is a unique equilibrium solution that gives the occupancies of the two states. This approach could be easily generalized to a Markov model with more than two states, where the transition between any two states is computed exactly as described above if that transition is deemed possible.

## Data visualization

All raw data is available for download and can be visualized with the help of the dms-view open-access tool [41]. On dms-view, it is possible to see the effects of different mutations for each residue of the Spike protein and visualise these on the 3D structure of Spike. Each analysed site is represented by 20 VDS values, one of them being zero (corresponding to the

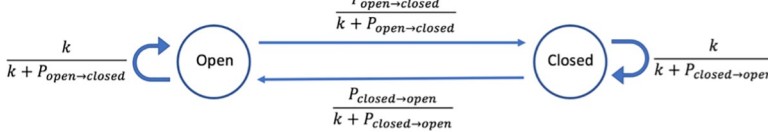

**Fig 2. Two-state Markov chain of Spike protein conformations.**

amino acid found in the wild type). The 'max' option will show the top VDS score for each position. Therefore, it shows which mutation for that specific position represents the candidate with the highest predicted infectivity as defined here in terms of a propensity to higher occupancy of the open state. The 'min' option will show the lowest score for each position and the mutation associated with the candidate predicted as least infectious. The 'median' option returns the median score, presenting a general trend for any given position, and 'var' shows the variance between the results for each position, highlighting sites in which mutations to different residues lead to a broader range of VDS values. Furthermore, for the mutations for which occupancy was calculated, the data can be accessed through the same menu. As new occupancy data is calculated, it will be added to this resource. Readers interested on the occupancy of particular mutations not yet available are invited to contact the authors via email or through the GitHub repository. When selecting each specific point on the first panel, it is possible to access all VDS values on the second panel and see the highlighted position in 3D on the structural representation.

## Results and discussion

### Dynamical Signature of different Spike variants

**Comparison of the differences in dynamics between G614 and D614.** An important event in the progression of the COVID-19 pandemic was the appearance of the D614G variant in mid-February 2020 in Europe. The fast spread of this variant raised the possibility that this mutation conferred advantages relative to other forms of the virus in circulation at the time [42,43]. Studies revealed that the mutation has indeed greater infectivity, triggering higher viral loads [44,45]. Several hypotheses have emerged to explain the mechanisms behind this higher infectivity, focusing primarily on possible effects on the Furin cleavage site [30,46,47], but recently also considering possible important dynamic differences [45,48,49].

In order to test if Dynamical Signatures reveal differences between Spike variants, we analysed the 13741 sequences of the protein available on May 8[th], 2020 in the COVID-19 Viral Genome Analysis Pipeline, enabled by data from GISAID [50,51]. The mutant Spike proteins harboring mutations (S1 Table) were modelled in the open and closed states. Dynamical Signatures were calculated for each mutant in both states and clustered (Fig 3) using the Euclidean distance between Dynamical Signature vectors as measure of dissimilarity. Mutations in positions that had no occupancy in the original templates used for the open and closed states (positions 5, 8, and 1263) were ignored.

Analysis of the effect of mutations on the Dynamical Signature shows that the D614G mutation produces similar dynamical patterns largely independent of the other mutations accumulated, and dynamical patterns that are distinct from that of the wild type and other mutants on both the open and closed states, as highlighted in the sections of the dendrogram marked in red. The dynamical characteristics of D614G are very specific and cannot be obtained with random mutations (S1 Fig and S2 Table).

When checking the difference between the Dynamical Signatures of the wild type D614 and the mutant G614 we observe that for the closed conformation, the pattern tends towards negative values, indicating that this mutation makes the closed state more flexible, especially around the position of the mutation. On the other hand, for the open B chain conformation the pattern is positive for the open RBD, the same chain NTD and the adjacent chain NTD, indicating that this mutation makes these areas of the open conformation more rigid (Fig 4).

This result led us to hypothesize that a more flexible closed state would favor the opening of Spike and that a more rigid open state would disfavor its closing, thus shifting the conformational equilibrium towards the open state and favouring interaction with ACE2, leading to

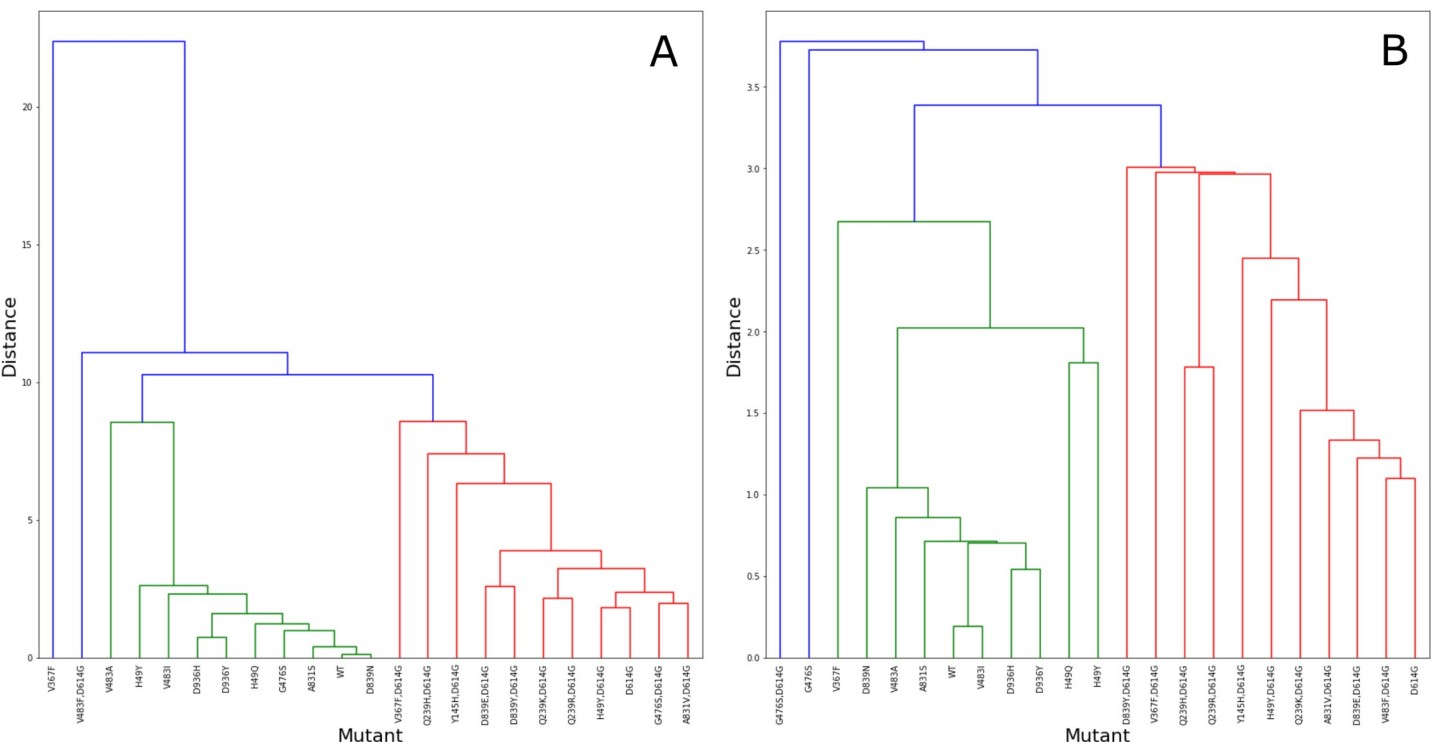

**Fig 3.** Dynamical Signature clustering for the closed (A) and open (B) state structures for WT and 22 mutants from GISAID (S1 Table). The clustering measures the distance between each pair of Dynamical Signature. Different colors were used to identify branches within a threshold of similarity (8.7 and 3.2 for closed and open state, respectively). The branches that comprehend most strains containing the D614G mutation are highlighted in red.

increased cell entry. Mutating position 614 to every other amino acid, we observe a correlation in the closed state between residue size and flexibility. Namely, smaller amino acids tend to make the closed state more flexible. However, we do not observe the opposite effect on the open state. Mutation of D614 to glutamine, which is similar to aspartate, barely shows any effect. Nevertheless, we can see that other amino acids have a similar effect as glycine, such as proline and threonine (S2 Fig).

**Comparison of the Dynamical Signatures of Spike from SARS-CoV and SARS-CoV-2.**
It has been previously observed that RBD flexibility in SARS-CoV influences binding to ACE2 and facilitates fusion with host cells [52]. Thus, considering the lesser infectivity of SARS-CoV relative to SARS-CoV-2 and our aforementioned results for the D614G mutation, we expected the SARS-CoV Spike to be more rigid in the closed state and more flexible in the open state relative to Spike from SARS-CoV-2. This is indeed the case (Fig 5). The Dynamical Signature values of SARS-CoV are smaller than those of SARS-CoV-2 in several areas throughout the closed structure, indicating that when in the closed state, the SARS-CoV Spike protein is more rigid. For the open state we can see that SARS-CoV open RBD and adjacent NTD are significantly more flexible than for SARS-CoV-2 Spike.

## Vibrational entropy

It is possible to combine the trend of a Dynamical Signature into a single value to represent the overall flexibility of any given mutation and compare it to the WT. This can be achieved with $\Delta S_{vib}$, calculated with Eq 2 for each state (see Materials and Methods). For any given state, positive $\Delta S_{vib}$ values represent mutants that relative to the wild type make the protein more rigid,

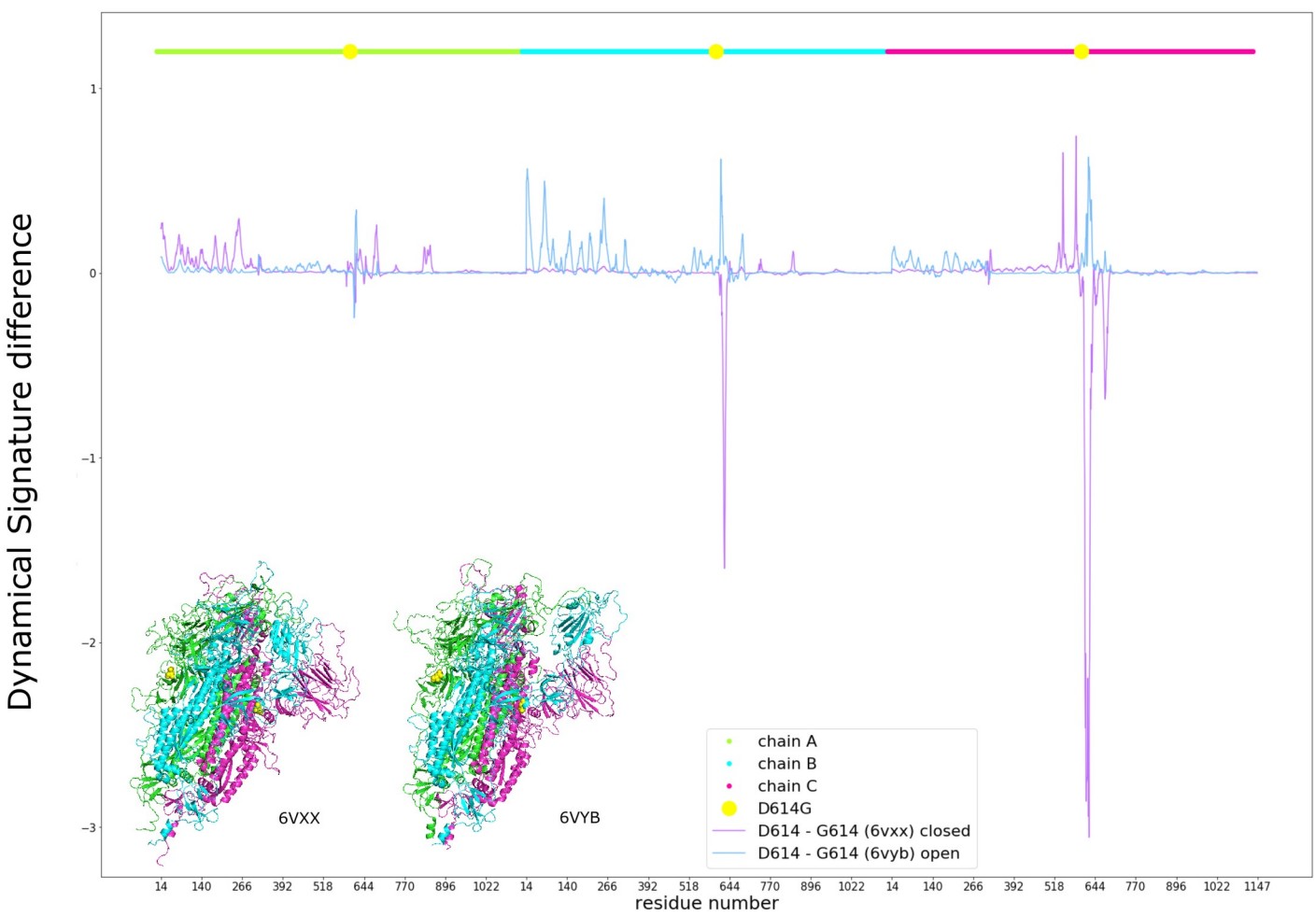

**Fig 4. Effects of the D614G mutation on the Dynamical Signature of the closed (purple) and open B chain RBD (blue) structures, measured by the difference between the calculated b-factors of D614 and G614.** Chains are represented in different colours and the position of the mutation is marked in yellow, using the same colours as for different regions of the structure as represented in the colours of the structures.

whereas negative values of $\Delta S_{vib}$ describe mutations that cause the protein to be more flexible in the given state relative to the wild type. In the case of the mutation D614G, we obtain $\Delta S_{vib\ (open)} = 5.26 \times 10^{-2}$ J.K$^{-1}$ and $\Delta S_{vib\ (closed)} = -9.27 \times 10^{-2}$ J.K$^{-1}$ with a VDS (calculated as $\Delta S_{vib\ (open)} - \Delta S_{vib\ (closed)}$) of $1.45 \times 10^{-1}$ J.K$^{-1}$.

We generated *in silico* the 19 possible single mutations in each position from residue 14 to residue 913 and calculated $\Delta S_{vib\ (open)}$, $\Delta S_{vib\ (closed)}$ and VDS. Other positions were ignored due to uncertainties in modelling or the fact that they are not expected to have a pronounced effect on dynamics [23]. It should be noted that Spike cannot accommodate the vast majority of such single mutations, particularly in its core as these would lead to unstable or misfolded conformations. However, those that occur near the surface are more likely to represent single residue variations of the Spike protein that lead to a stable, correctly folded protein. Therefore, the stability of specific mutations highlighted in this work, unless otherwise stated (such as those already observed experimentally or within the RBD domain as stated below), needs to be validated experimentally.

The heatmap in Fig 6A shows $\Delta S_{vib}$ values associated with mutations on the closed conformational state (left) and open conformational state (right). Lighter colors represent high $\Delta S_{vib}$

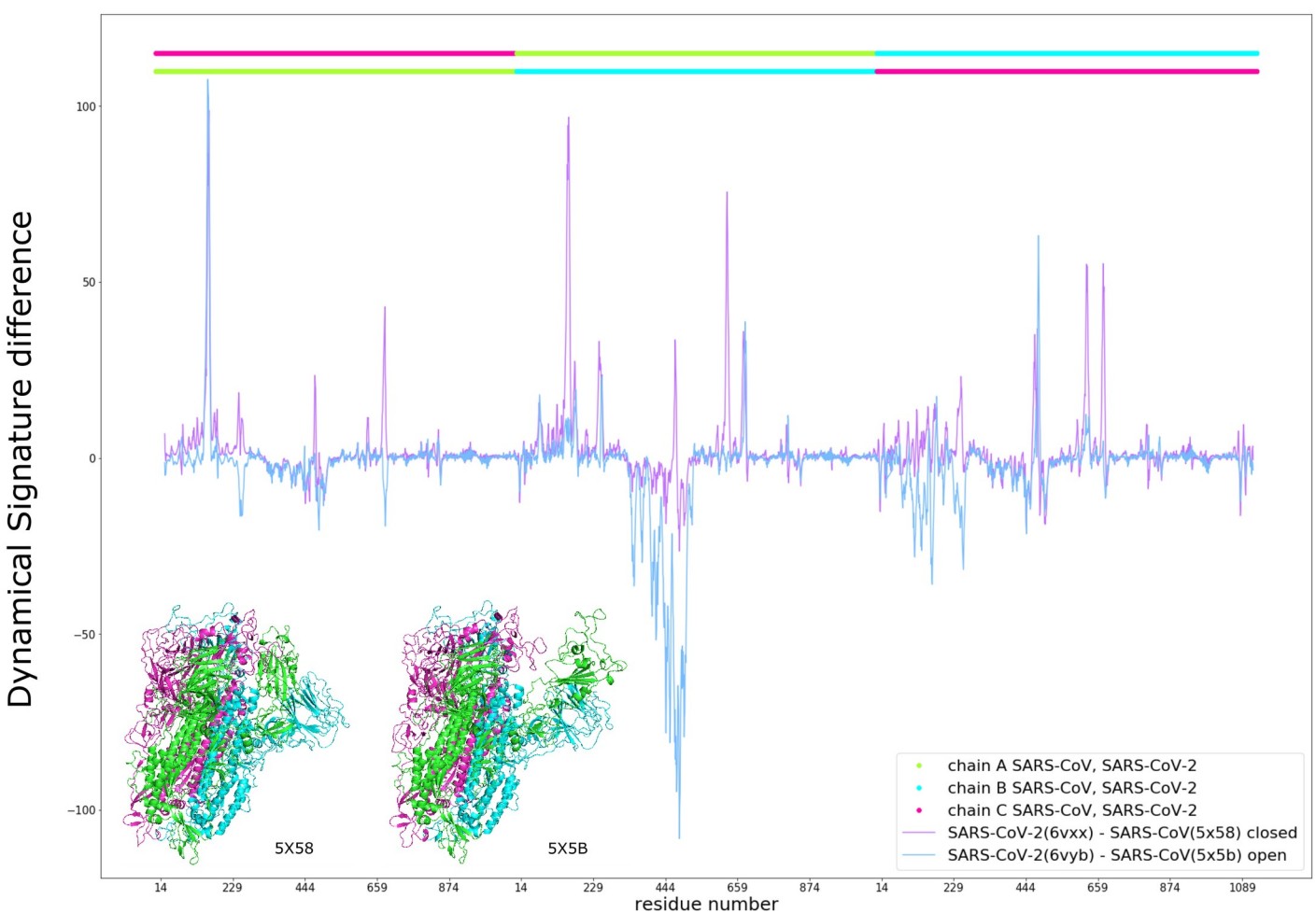

**Fig 5. Comparison between SARS-CoV-2 and SARS-CoV.** Dynamical Signature difference of the closed (purple) and open B chain RBD (blue) between aligned residues of the Spike protein from SARS-CoV-2 and SARS-CoV, with SARS-CoV chains represented in the top bar and equivalent colors in the structures and SARS-CoV-2 chains represented in the bar just below.

values, meaning that the specific mutant is more rigid than the WT, and darker colors represent low $\Delta S_{vib}$ values, meaning that the specific mutant is more flexible than the WT. The second heatmap (Fig 6B) shows VDS values, highlighting positions and specific mutations with great contrast between their effect on the open and closed states. In this representation, blue mutants are more rigid in the closed state and more flexible in the open state, therefore candidates for less infectious mutants, and red mutants are more flexible when closed and more rigid when open, candidates for more infectious mutants.

In Fig 7 we map VDS values (Fig 6B) on the structure of Spike, colored according to the median value for each position with the same color scheme as the heatmap. From the 17081 single mutations considered, we show the top 64 mutants with VDS>3.00x10$^{-1}$ J.K$^{-1}$ (Tables 1 and S3) as well as the bottom 20 in terms of VDS values (S3 Table). The mutants with predicted open state occupancy higher than that of the wild type are presented in Table 1. The Dynamical Signature comparison for 3 of those most infectious candidates (S3A Fig) and 3 of the least infectious candidates (S3B Fig) shows some of the patterns that could lead to a greater or lesser predicted effect on infectivity. For instance, in Figs 8 and S3A we can see that high scores can come from a large flexibility of the closed state, a very large rigidity of the open

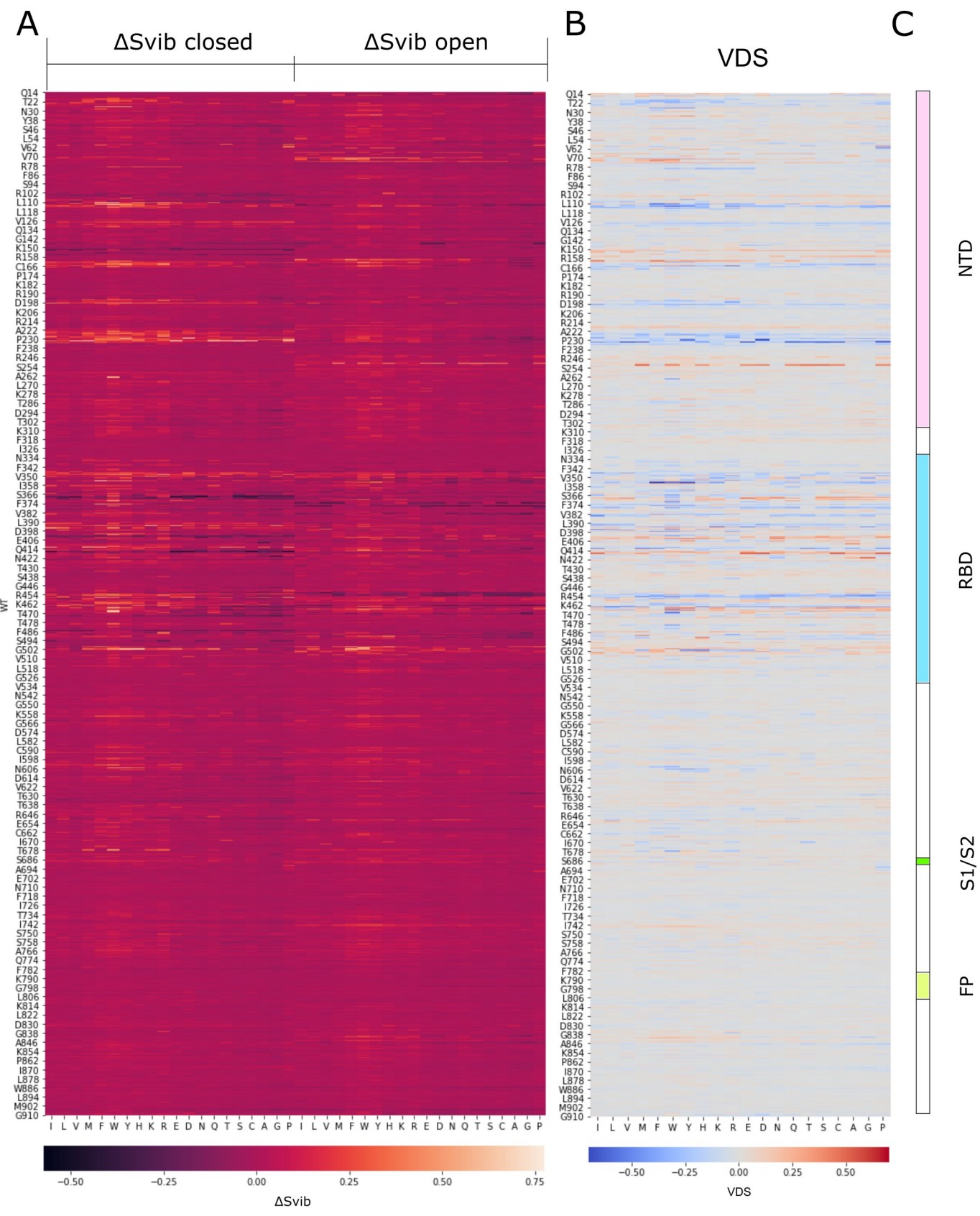

**Fig 6.** Heatmaps representing the values of $\Delta S_{vib}$ for the closed structure (A, left-hand section) and for the open structure (A, right-hand section), and the values of VDS (B) for every possible mutant in Spike from positions 14 to 913. Each column represents one of the 20 amino acids (repeated in the left heatmap). Notice that for each position (represented in a row), one particular column represents the value of the WT amino acid found at that position. Higher values of $\Delta S_{vib}$ are represented in yellow and lower values in dark purple. Higher values of VDS are represented in red and lower values in blue. The domain structure of Spike is represented in (C) for reference purposes.

state, or have the contribution of both. We can also observe that these effects can be different in each chain and can affect more the NTD, the RBD, or both. Finally, these single mutants also show how a point mutation can have widespread impacts on flexibility across the whole protein.

## Conformational state occupancies

We calculated forward and reverse transition probabilities between the open and closed states (Eq 4, 5 & 6) from the calculated normal modes and used the Markov model described in Materials and Methods to calculate the equilibrium occupancies for each state in wild type and mutant Spike proteins. It is unclear if any additional conformational states other than those with either all three RBD domains in the closed state or only one RBD open state are biologically relevant. Specifically, Yurkovetskiy *et al.* [45] observed an occupancy for states with two or three RBD domains in the open conformation, but these were not observed by Gobeil *et al.* [30] and Xiong *et al.* [34] or taken into consideration in several other structural studies [20–24]. As such, we employ the two-state model shown in Fig 2, with one state representing all three RBD domains closed and the second state representing one RBD open.

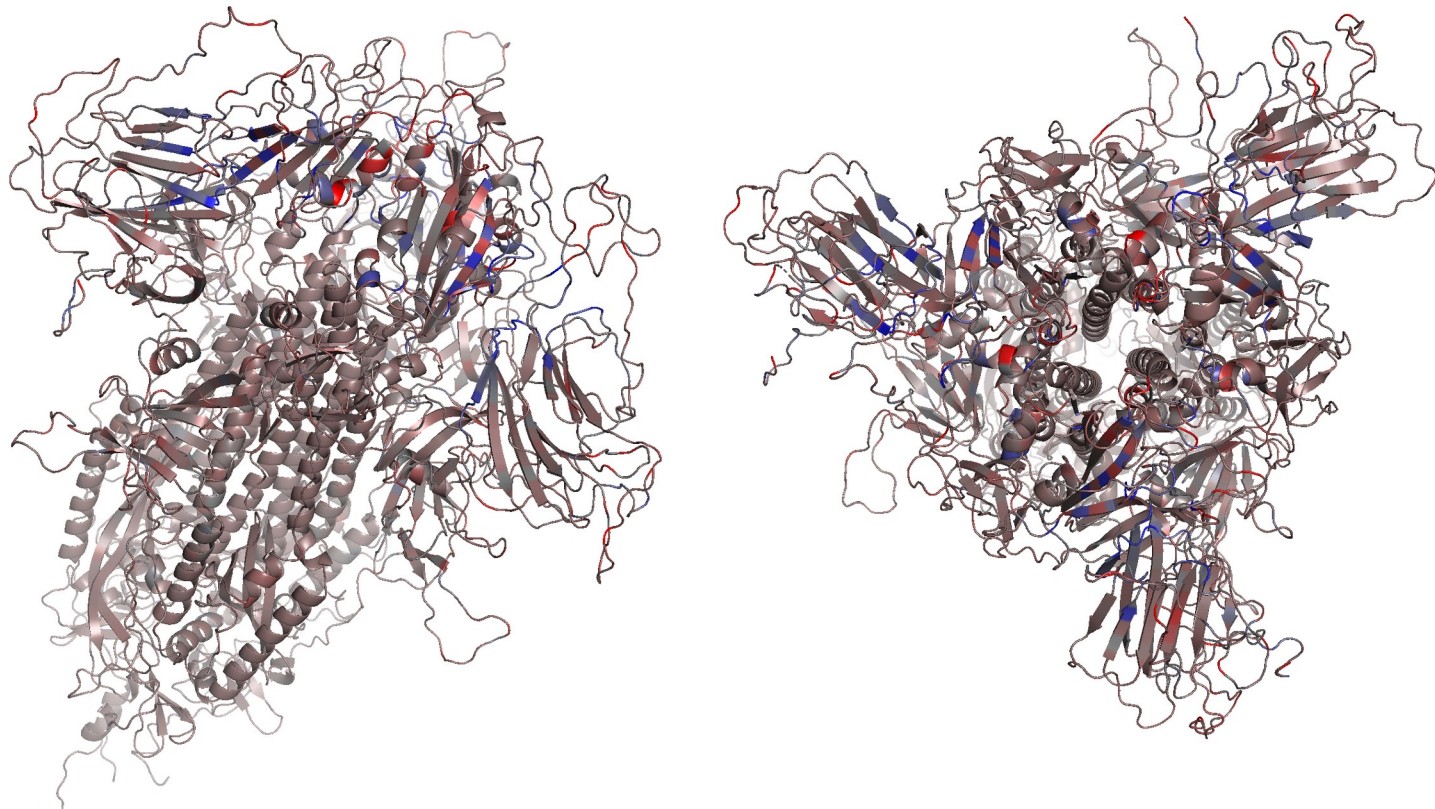

**Fig 7. VDS values represented in the structure of Spike from two angles according to the median value for each position and the same color scheme as in the Difference Score Heatmap (Fig 6B).**

**Table 1. Putative mutations, associated VDS ($\Delta S_{vib\ (open)} - \Delta S_{vib\ (close)}$, in units of J.K$^{-1}$) and predicted occupancies for the open and closed states for the mutants with predicted open-state occupancy higher than that of the wild type.** Predicted occupancy values are shown for the open conformation, the closed conformation, and the difference between the two (closed–open). The data for the remaining mutants with occupancy below that of the wild type but VDS>0.3 (red) as well as those with the lowest predicted VDS values (blue) is presented in S3 Table.

| Variant | VDS | Predicted Occupancy | | |
|---|---|---|---|---|
| | | Open state | Closed State | Difference (Open–Closed) |
| N501W | 0.372 | 62.705% | 37.295% | 25.410% |
| G504I | 0.349 | 40.491% | 59.509% | -19.019% |
| G416E | 0.314 | 30.111% | 69.889% | -39.778% |
| G416Y | 0.403 | 28.478% | 71.522% | -43.045% |
| R403S | 0.323 | 26.794% | 73.206% | -46.412% |
| D467W | 0.431 | 26.379% | 73.621% | -47.243% |
| G252W | 0.383 | 26.302% | 73.698% | -47.396% |
| G404W | 0.387 | 26.302% | 73.698% | -47.396% |
| G252Q | 0.394 | 26.295% | 73.705% | -47.410% |
| G252H | 0.353 | 26.268% | 73.732% | -47.464% |
| G252E | 0.365 | 26.224% | 73.776% | -47.551% |
| K417G | 0.315 | 26.179% | 73.821% | -47.641% |
| G252C | 0.401 | 26.164% | 73.836% | -47.672% |
| G252S | 0.447 | 26.159% | 73.841% | -47.682% |
| G252T | 0.440 | 26.157% | 73.843% | -47.686% |
| G252D | 0.390 | 26.149% | 73.851% | -47.702% |
| G413M | -0.521 | 26.147% | 73.853% | -47.705% |
| G252M | 0.481 | 26.134% | 73.866% | -47.731% |
| G252P | 0.403 | 26.120% | 73.880% | -47.759% |
| K417D | 0.518 | 26.114% | 73.886% | -47.773% |
| D467Y | 0.399 | 26.106% | 73.894% | -47.788% |
| S161F | 0.395 | 26.056% | 73.944% | -47.888% |
| K417P | 0.502 | 26.035% | 73.965% | -47.930% |
| S161Y | 0.303 | 25.980% | 74.020% | -48.040% |
| K417E | 0.494 | 25.971% | 74.029% | -48.058% |
| D467P | 0.308 | 25.954% | 74.046% | -48.093% |
| R34Y | 0.380 | 25.908% | 74.092% | -48.184% |
| I468T | 0.350 | 25.907% | 74.093% | -48.186% |
| R355F | -0.697 | 25.889% | 74.111% | -48.222% |
| S161I | 0.391 | 25.855% | 74.145% | -48.290% |
| G72W | 0.404 | 25.851% | 74.149% | -48.298% |
| T73F | 0.376 | 25.844% | 74.156% | -48.313% |
| Q14C | 0.312 | 25.839% | 74.161% | -48.322% |
| WT | 0.000 | 25.837% | 74.163% | -48.327% |

The Markov model calculation of occupancies requires two parameters (see Materials and Methods) that were optimized based on experimental data for six Spike variants. These variants were: S-GSAS/D614, an engineered Spike with the sequence GSAS in the Furin cleavage site and no 614 mutation; S-GSAS/G614, with the same Furin site modifications and the D614G mutation [30]; S-R, the Spike protein with original Furin site RRAR; S-R/x2, with added S383C, D985C mutations inducing a disulfide bond; S-R/PP, engineered with two prolines in positions 986 and 987; S-R/PP/x1, in which from the double prolines sequence the mutations G413C, V987C were performed to induce a disulfide bond [34]. It is worth stressing that all 6 variants used to calibrate the two parameters affecting the occupancy were modelled

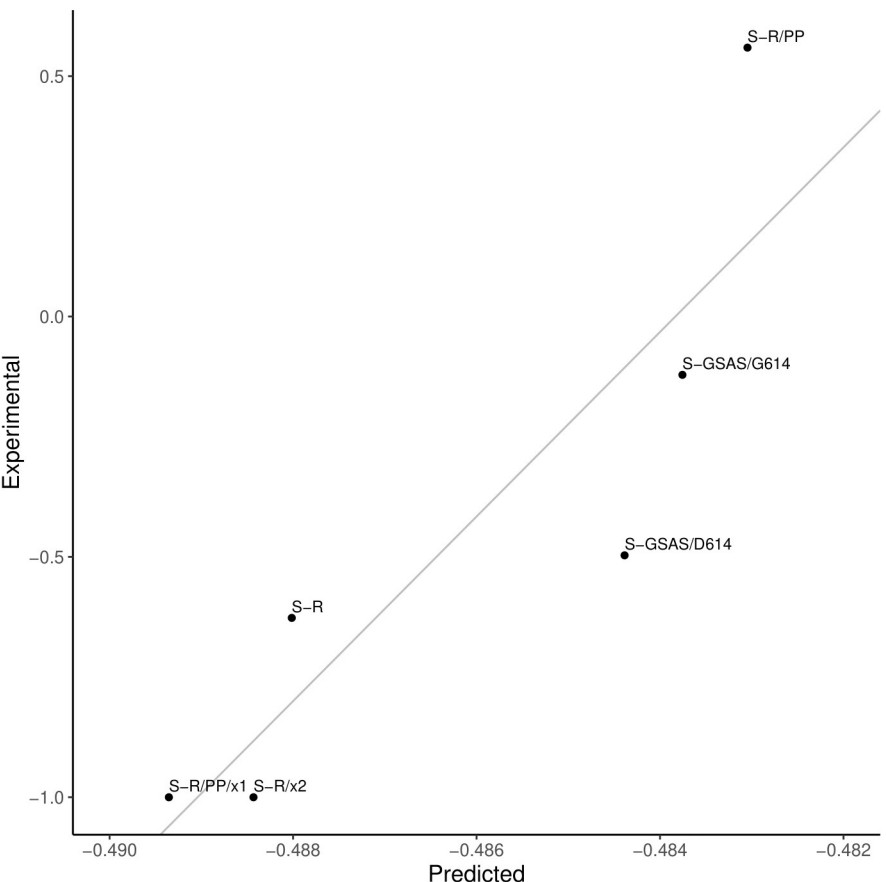

**Fig 8. Difference in the occupancies for the open and the closed states (open–closed) for six variants of the Spike protein.** Experimental values are represented on the Y-axis and the predicted values in the scale on the X-axis. Predicted values for the parameters k = 0.5, γ = 0.001. Represented linear fit of *Experimental = 192.011\* Predicted + 92.9013*. Errors on the experimental measurements are not known.

on the same open and closed state conformations. All differences in observed occupancies and the agreement with experimental occupancy data came about as a direct consequence of the effect of the mutations on the normal modes and derived transition probabilities and not as a result of structural differences between variants. We obtained a good fitting to the experimental results with k and γ of 0.5 and 0.001, respectively (Pearson correlation = 0.89, p-value = $1.94 \times 10^{-2}$). Predicted occupancies of the open and closed states for each of the six variants above, as well as the experimental data, are presented in Table 2.

**Table 2. Experimental and predicted occupancies for the open and closed states and their difference for multiple SARS-CoV-2 variants.** Experimental values obtained from Gobeil *et al.* [30] and Xiong *et al.* [34].

| Variant | Experimental Occupancy | | | Predicted Occupancy | | |
|---|---|---|---|---|---|---|
| | **Open** | **Closed** | **difference** | **Open** | **Closed** | **difference** |
| S-GSAS/D614 | 25.160% | 74.840% | -49.680% | 25.781% | 74.219% | -48.439% |
| S-GSAS/G614 | 43.938% | 56.062% | -12.123% | 25.812% | 74.188% | -48.376% |
| S-R/x2 | 0.000% | 100.000% | -100.000% | 25.578% | 74.422% | -48.843% |
| S-R/PP/x1 | 0.000% | 100.000% | -100.000% | 25.532% | 74.468% | -48.935% |
| S-R | 18.646% | 81.354% | -62.707% | 25.599% | 74.401% | -48.801% |
| S-R/PP | 77.967% | 22.033% | 55.935% | 25.848% | 74.152% | -48.305% |

We utilized these data to calculate occupancy differences for each variant (Fig 8). The range of variation of our predicted occupancies is small compared to that of experimental values. We believe that given the limitations of our coarse-grained model as well as additional phenomena that ultimately affect occupancy, our predictions reflect only a fraction of the myriad of factors contributing to the occupancy. Nonetheless, our predictions correctly capture the pattern of relative variations of occupancy observed in the experimental data. To ensure that the calculated correlation is not due to chance, we simulated random sets of occupancies for the 6 sequence variants and calculated simulated correlations for the 110 different combinations of k and γ to determine if the observed correlations represent an actual signal in the data or could be randomly obtained with different values for the parameters k and γ. We observed a marked shift with higher correlations for the data representing our predicted occupancies when compared to the gaussian noise data (S4 Fig), suggesting that the predicted occupancies are not due to chance.

We set a threshold of VDS>0.3 to select candidates for the calculation of occupancies, corresponding to 64 mutations (Tables 1 and S3, in red). Using the parameters k and γ obtained above, we calculated occupancies for these 64 mutants as well as the 20 mutants with lowest VDS values (Tables 1 and S3, in blue) for comparison. In Fig 9A we show the difference in occupancy between the open and closed states using a non-linear scale adapted to better show the results around the wild type occupancy. Whereas VDS values for particular mutations may hint at a more flexible closed state and more rigid open state, this is a global measure that may not reflect the necessary pattern of flexibility across the structure that leads to effective transition probabilities between the open and closed states. Yet, for the most part, VDS can predict the shifts in occupancy, showing a clear distinction between the 64 mutants predicted using VDS as shifting occupancy towards the open state and the 20 mutants predicted to shift the equilibrium towards the closed state (p-value = 2.04x10$^{-6}$). Fig 9B shows the location in the structure of the mutants in Table 1. We can see that the least infectious candidates (blue) are positioned in the interfaces between NTD and RBD domains, while the most infectious candidates, especially the ones validated by the occupancy prediction (dark red), are more concentrated in the interfaces between different RBD domains.

Residue G252 stands out as capable of accommodating a large number of mutations (C, D, E, H, M, P, Q, S, T, W) that shift the occupancy in favour of the open state. The fact that variants in this position do not seem to be prevalent in outbreaks to date, points to the possibility that this position may be under additional functional constraints that prevent the emergence of variants. A number of other glycine residues could also accept mutations that we predict to increase the occupancy of the open state: G72W; G404W; G413M; G416E,W; and G404I. In fact, three of the top four mutations are mutations on glycines. A number of other potential mutations are adjacent to the glycine residues above. Namely, R403S and K417D,E,G,P. Additionally, D467P,W and I468T are also positions that are adjacent to others that can accommodate mutations that may lead to a conformational shift favouring the open state. The mutation that favours the open state the most in our calculations is N501W with $\Delta S_{vib\ (open)} = 6.02x10^{-1}$ J.K$^{-1}$ and $\Delta S_{vib\ (closed)} = 2.30x10^{-1}$ J.K$^{-1}$ and a resulting VDS value of $3.72x10^{-1}$ J.K$^{-1}$ leading to occupancies compared to those of the wild type (in parenthesis) of 62.7% (25.8%) and 37.3% (74.2%) for the open and closed states respectively. It is important to stress, as discussed in Materials and Methods, that the calculations are performed using structures containing a modified Furin recognition site and prolines in positions 986 and 987. Furthermore, the contribution of vibrational entropy changes is one among potentially several effects, the overall importance of which remains to be determined. Therefore, relative changes in occupancy are relevant whereas the specific values are less so.

The COG-UK consortium (https://www.cogconsortium.uk/about/) monitors the appearance and spread of new strains of SARS-CoV-2. COG-UK recently detected a strain containing

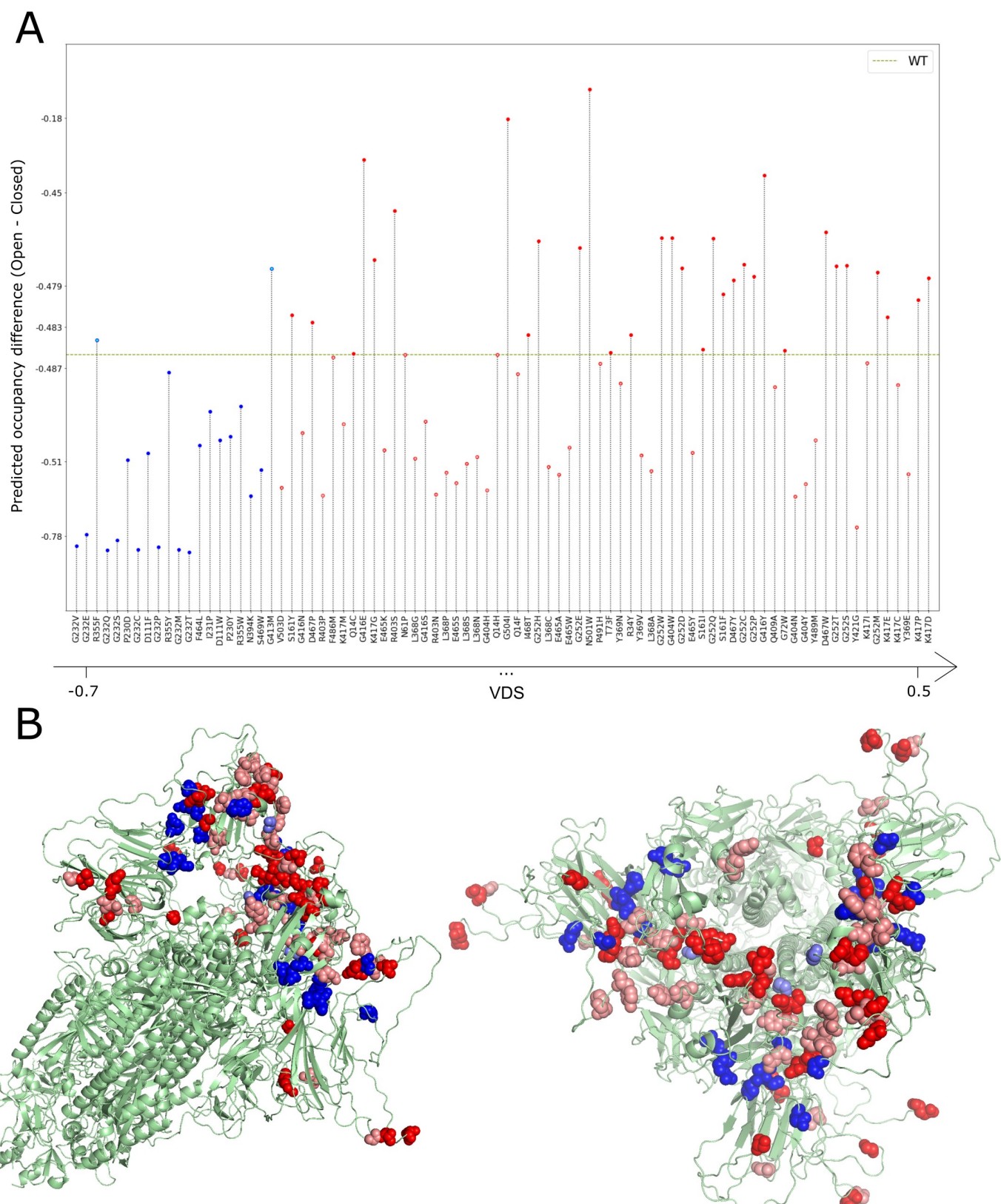

**Fig 9.** (A) Difference in the occupancies for the open and the closed states for the top 64 mutants with VDS>0.3 (red) and the 20 mutants with lower VDS (blue). Occupancy difference for the WT is represented by the dashed green line. Y-axis based on the transformation of a symmetric logarithmic scale. (B) Two

visualizations of the 6VYB structure highlighting the mutations. The bottom 20 mutant positions are marked in two shades of blue, with the darker shade indicating positions in which at least one mutant had an (open–closed) occupancy value smaller than wild type. The top 64 mutant positions are marked in two shades of red, with the darker shade indicating positions in which at least one mutant had an (open–closed) occupancy value higher than wild type.

the mutation N501Y that has been observed to be spreading rapidly at the time of writing. We believe that shifts in occupancy may be in part responsible for its emergence. According to our calculations, the N501Y mutant shows $\Delta S_{vib\,(open)} = -1.60 \times 10^{-2}$ J.K$^{-1}$ and $\Delta S_{vib\,(closed)} = 2.37 \times 10^{-1}$ J.K$^{-1}$, with VDS = $2.53 \times 10^{-1}$ J.K$^{-1}$. The predicted occupancies for the N501Y mutant compared to those of the wild type (in parenthesis) are 54.3% (25.8%) and 45.7% (74.2%) for the open and closed states, respectively. Therefore, the N501Y mutant shows a marked increase of the occupancy of the open state relative to other mutations. Additionally, this mutation was shown to also increase binding affinity to the ACE2 receptor relative to the wild type with a $\Delta\log_{10}(K_{D,app})$ of 0.24 [53]. Therefore, we predict that N501Y has a strong potential to contribute to increased transmission. The calculations above were performed in the context of D614. However, the double mutant representing the N501Y mutation in the context of G614 also shows an increase in the occupancy of the open state to 35.06%. The recently observed A222V mutation on the other hand [54], does not show in our analysis any propensity of altering the occupancy of states with a negative VDS of $-1.64 \times 10^{-2}$ J.K$^{-1}$. Predicted occupancies for A222 and V222 are nearly identical either in the context of D614 (WT) or the mutant containing G614.

Notice that N501Y has a VDS value of $2.53 \times 10^{-1}$ J.K$^{-1}$ that is slightly below the $3.00 \times 10^{-1}$ J.K$^{-1}$ threshold, suggesting that there may be many other mutations with VDS values below our set threshold that turn out to have augmented occupancies for the open state relative to the wild type.

D614G shows that changes in the occupancy of conformational states can impact infectivity despite no changes or even weaker binding affinities [45]. A recent study [53] on binding and expression of Spike mutations within the RBD domain (positions 331 to 531) shows that several (but not all, see below) of the mutations that we predicted to have increased occupancy of the open state are associated with a decrease of binding affinity with ACE2. Incidentally, the data also shows that the mutations in Table 1 within the RBD produce stable and properly folded Spike proteins. As shown for D614G, infection does not rely on binding affinity alone, and even a strain with higher dissociation rates from ACE2 can bring about fitness advantages.

The mutation N501W is predicted to have the largest effect in augmenting the occupancy of the open state relative to the wild type. This mutation is associated with stronger binding to ACE2 ($\Delta\log_{10}(K_{D,app})$ = 0.11) [53] relative to the wild type Spike (but lower than N501Y). Furthermore, N501W appears to have increased expression relative to the wild type with a $\Delta\log$ (MFI) of 0.1 compared to decrease in relative expression of -0.14 for N501Y [53]. The authors note that changes in expression correlate with folding stability [53]. However, even with a $\Delta\log$ (MFI) of -0.14, N501Y is viable and spreading. Therefore, N501W might be even more stable and infective.

We consider all mutations with increased predicted occupancy of the open state in Table 1 as good candidates for further experimental validation to better understand the role of binding and dynamics of Spike and their role in SARS-CoV-2 infectivity. Furthermore, we suggest that their appearance in outbreaks should be closely monitored.

## SARS-CoV-2 Variants: B.1.1.7, 501.V2, P.1, Delta and Delta+

The mutation N501Y above appears in the B.1.1.7 variant first observed in the UK [55] as well as the 501.V2 variant first observed in South Africa [56] and the P.1 variant from Brazil [57] that are rapidly spreading around the globe. These two strains contain additional mutations in

Spike. Namely, B.1.1.7 contains N501Y, A570D, D614G, P681H, T716I, S982A, D1118H and deletions on positions 69, 70 and 144. As the number of normal modes is related to the number of amino acids, we are unable to model deletions while still making comparisons with the wild type strain given the nature of the quantities calculated (Eqs 2 and 6). Therefore, the deletions of three residues at positions 69,70 and 144 that are present in B.1.1.7 were not modelled here. 501.V2 includes the mutations L18F, D80A, D215G, R246I, K417N, E484K, N501Y, D614G, A701V. P.1 variant includes the mutations L18F, T20N, P26S, D138Y, R190S, K417T, E484K, N501Y, D614G, H655Y, T1027I. The Dynamical Signatures for B.1.1.7, 501.V2 as well as P.1 show a strong rigidification of the open state and added flexibility of the closed state (S5–S7 Figs respectively), leading to VDS values of $5.30 \times 10^{-1}$ J.K$^{-1}$, $6.45 \times 10^{-1}$ J.K$^{-1}$ and $6.88 \times 10^{-1}$ J.K$^{-1}$ and open state occupancies of 36.2%, 35.8% and 39.8% for B.1.1.7, 501.V2 and P.1 respectively. The three variants show an increase in occupancy of approximately 40% relative to the wild type (25.8%). Despite our preference of modelling the smallest possible number of mutations and therefore using the engineered structure containing the modified Furin clivage site and proline modification, we also modelled B.1.1.7 (except the deletions), 501.V2 and P.1 using the original sequence of Spike with a non-modified Furin clivage site. We obtain 33.0%, 33.6%, and 38.9% occupancy for B.1.1.7, 501.V2 and P.1 respectively.

Different factors may contribute to the apparent evolutionary advantages of the above strains through the course of the pandemic. The N501Y substitution, as mentioned, is among the candidates we pointed as enabling occupancy shifts towards the open conformation, but was also shown to have higher binding affinity to the receptor ACE2 [53]. The E484K mutation does not increase the occupancy of the open state in our predictions and also does not increase ACE2 binding affinity [53], but was observed to facilitate immune escape in a study with human serum antibodies from subjects that recovered from COVID-19 [58]. However, mutations like K417N and K417T were not observed to cause favourable changes on expression (a proxy measurement for stability) or binding [53] as well as immune recognition [58], but in our high throughput evaluation they were, as well as several substitutions in this position, predicted to increase the occupancy of the open state.

All predictive studies on Spike mutations, with both computational and experimental approaches, covered single mutations only. Therefore, it is possible that combinations of substitutions that constitute each new variant bring about advantages due to a number of factors that are hard to decouple or are yet to be determined. For example, the Indian variant B.1.617.2 (also known as Delta) [59], contains a core number of mutations in Spike (G142D, E154K, L452R, E484Q, D614G, P681R, Q1071H) but also additional sub-strain variations (T95I, H1101D or V382L) [60]. The B.1.617.2 strain and its variations above have predicted open state occupancies comparable or lesser than that of the wild type in our calculations. This suggests that other unknown factors alone or in combination are at play if the preliminary data suggesting increased transmissibility [61] is observed on larger samples. A newer variation of the Delta strain, AY.1 (or Delta+) does however contain the K417N mutation (that we predict to increase open state occupancy) similar to the South African strain 501.V2. A preliminary report [62] places AY.1 as a variant of concern (VOC).

## Conclusions

SARS-CoV-2 mutations are still arising and spreading around the world. The A222V mutation, reportedly responsible for many infections, emerged in Spain during the Summer of 2020 and since then has spread to neighbor countries [54]; In Denmark, new strains related to SARS-CoV-2 transmission in mink farms were confirmed in early October by the WHO and shown to be caused by specific mutations not previously observed with the novelty of back-

and-forth transmission between minks and humans [63]. A new strain containing N501Y first appeared in the UK, the recent Delta+ strain (AY.1) containing K417N is now on the rise at the time of writing. Such occurrences point to the possibility that new mutations in SARS-CoV-2 may bring about more infectious strains.

Using the low computational cost methods described in this paper, it is possible to predict potential variants that might have an advantage over the wild type virus insofar as these are the result of changes in occupancy of states. Despite the limitations of the simplified coarse-grained model employed here, our results correctly model the experimentally observed higher open state occupancy of several mutations. In our analyses, flexibility properties and conformational state occupancy probabilities contribute to the infectivity of a SARS-CoV-2. Our results explain the behaviour of the D614G strain, the increased infectivity of SARS-CoV-2 relative to SARS-CoV as well as offer a possible explanation for the rise of new strains such as those harboring the N501Y mutation.

The results we present on SARS-CoV-2 Spike mutations have several limitations. First and foremost, some of the *in silico* mutations discussed may not be thermodynamically stable, may affect expression, cleavage, or binding to ACE2, and our approach does not consider that Spike is, in fact, a glycoprotein and the sugar molecules may have an effect on dynamics. However, the agreement between our model and experimental observations as described above shows that the simplified model of Spike and the coarse-grained methods used here allow us to calculate dynamic properties of Spike that are relevant to understand infection and epidemiological behavior. It is important to keep in mind that all of the mutations that we discuss in Table 1 that lay within positions 331 and 531 within the RBD domain were already experimentally validated shown to be viable [53]. However, we highlight the need for experimental validation of our predictions particularly for those candidates that we believe would help elucidate the extent of the effect of the conformational dynamics of Spike on infectivity. Beyond *in vitro* biophysical studies, experimental alternatives exist such as using pseudo-type viruses or virus-like-particles that would not require studying gain-of-function mutations using intact viruses. Alternatively, loss-of-function mutations can be created with intact viruses and compared to the wild type SARS-CoV-2 virus to validate the role of dynamics on infectivity.

After this work first became available as a preprint in December 2020 [64], extensive molecular dynamics simulations totalling 20 μs and investigating the effect of the D614G mutation on conformational dynamics have been conducted with the full complement of glycans [65]. The authors found that this mutation increased the occupancy of the up (open) state, in line with the results presented in our work. This makes us confident that our coarse-grained model lacking glycans still captures the essential dynamics of the Spike protein and validates its application for high throughput screening of mutations, which would be too costly computationally to conduct using long all-atom MD simulations.

Two studies determined the structures of the D614G mutant [66,67] in open and closed states via cryo electron microscopy (cryo-EM) in the open and closed states were published and came to our attention after this manuscript was published as a preprint. Both studies come to the same conclusion as ours, namely that the open state is favoured on the D614G mutant. More recently, the structures of the B.1.1.7 (UK), 501.V2 (SA) and P1 (Brazil) variants modelled here become available [68]. The authors show in all cases an increased propensity for the open state. In summary, while this computational work has been under review a number of studies corroborating our conclusions have appeared. Namely, complementary computational molecular dynamics simulations as well as experimental work for the single mutant D614G as well as experimental work on the UK, SA and Brazil variants.

This work demonstrates how a simplified coarse-grained model can capture essential aspects of the effect of mutations on the dynamics between conformational states for the

SARS-CoV-2 Spike protein. Furthermore, the low computational cost of the calculations allowed us to predict mutations with evolutionary advantages that appeared during the COVID-19 pandemic. Lastly, as the COVID-19 pandemic is still ongoing, our high-through-put data can contribute to the risk assessment of future variants.

To the best of our knowledge, this is the first time that a normal mode analysis method is used to model the effect of mutations on the occupancy of conformational states, opening a new opportunity in computational biophysics to create dynamic models of transitions between conformational states of proteins based on physical properties and sensitive to sequence variations. We hope that our results contribute to inform the research community in understanding SARS-CoV-2 infection mechanisms, open new possibilities in computational biophysics to study protein dynamics and help public health surveillance programs decide on the risk posed by new strains given the appropriateness of our method for large-scale uses.

## Supporting information

**S1 Fig. Dynamical Signature clustering between WT, 22 mutants observed in nature and 30 random mutants designed, both for the full structure (A) and only for the RBD (B). The grey arrow highlights the clusters containing most of the mutants that have the mutation D614G.**
(TIF)

**S2 Fig. Effects on the Dynamical Signatures of the mutation from D614 to each one of the other 18 possible residues.**
(TIF)

**S3 Fig. Dynamical Signature differences for three mutations among the top VDS (A)– K417E, G252M and G404Y –and bottom VDS values (B)–R355Y, F464L and I231P –for closed (purple) and open (blue) B chain RBD structures. Chains A, B and C are marked on the top in green, cyan and magenta, respectively, and the point mutations are marked in yellow.**
(TIF)

**S4 Fig. Simulated Pearson correlations between gaussian noise vectors of length 6 for the 110 iterations associated to each combination of the parameters k = [0.001, 0.005, 0.01, 0.05, 0.1, 0.5, 1, 5, 10, 50, 100] and $\gamma$ = [0.001, 0.01, 0.1, 1, 10, 100, 1000, 10000, 100000, 1000000].**
(TIF)

**S5 Fig. Dynamical signature difference for the open (blue) and closed (purple) states relative to the wild type for the B.1.1.7 variant.**
(TIF)

**S6 Fig. Dynamical signature difference for the open (blue) and closed (purple) states relative to the wild type for the 501.V2 variant.**
(TIF)

**S7 Fig. Dynamical signature difference for the open (blue) and closed (purple) states relative to the wild type for the P.1 variant.**
(TIF)

**S1 Table. Amino acid mutations associated to 13741 sequences of the Spike protein available on May 08 in COVID-19 Viral Genome Analysis Pipeline, enabled by data from**

**GISAID.**
(DOCX)

**S2 Table. Random mutants accumulating from one to four mutations.**
(DOCX)

**S3 Table. Putative mutations and associated VDS ($\Delta S_{vib\,(open)} - \Delta S_{vib\,(close)}$, in units of J.K$^{-1}$) and predicted occupancies with open occupancies lower than for the wild type for the 64 mutants with FDS>0.3 and the 20 mutants with lowest FDS values.**
(DOCX)

## Acknowledgments

RN is a Fonds de Recherche du Québec—Santé (FRQ-S) Senior Fellow, a member of the Réseau Québécois de Recherche sur les Médicaments (RQRM) and the Quebec Network for Research on Protein Function, Engineering and Applications (PROTEO). The authors would like to dedicate this work to the memory of Mordechai Najmanovich, Z"L, father of RN, who passed away from complications due to COVID-19 on November 26, 2020. RN would like to thank all healthcare workers, particularly ICU nurses and physicians at the Avista Adventist Hospital in Louisville, Colorado, for their efforts.

## Author Contributions

**Conceptualization:** Natália Teruel, Olivier Mailhot, Rafael J. Najmanovich.

**Data curation:** Natália Teruel, Olivier Mailhot.

**Formal analysis:** Natália Teruel, Olivier Mailhot, Rafael J. Najmanovich.

**Funding acquisition:** Rafael J. Najmanovich.

**Investigation:** Natália Teruel, Olivier Mailhot, Rafael J. Najmanovich.

**Methodology:** Natália Teruel, Olivier Mailhot, Rafael J. Najmanovich.

**Project administration:** Rafael J. Najmanovich.

**Resources:** Olivier Mailhot.

**Software:** Natália Teruel, Olivier Mailhot.

**Supervision:** Rafael J. Najmanovich.

**Validation:** Natália Teruel, Olivier Mailhot.

**Visualization:** Natália Teruel, Olivier Mailhot.

**Writing – original draft:** Natália Teruel, Olivier Mailhot.

**Writing – review & editing:** Natália Teruel, Olivier Mailhot, Rafael J. Najmanovich.

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
