## [Decision Letter · Decision Letter 0]

14 May 2021

Dear Dr. Najmanovich,

Thank you very much for submitting your manuscript "Modelling conformational state dynamics and its role on infection for SARS-CoV-2 Spike protein variants" for consideration at PLOS Computational Biology.

As with all papers reviewed by the journal, your manuscript was reviewed by members of the editorial board and by several independent reviewers. In light of the reviews (below this email), we would like to invite the resubmission of a significantly-revised version that takes into account the reviewers' comments.

Both reviewers were positive overall. The first reviewer had concerns primarily  about the presentation. However, the second reviewer had more significant concerns about some of the methodology (glycans) and focused comparison to experimental data, which I think would improve the paper.

We cannot make any decision about publication until we have seen the revised manuscript and your response to the reviewers' comments. Your revised manuscript is also likely to be sent to reviewers for further evaluation.

Sincerely,

Roland L. Dunbrack Jr., Ph.D.

Associate Editor

PLOS Computational Biology

Arne Elofsson

Deputy Editor

PLOS Computational Biology

Reviewer's Responses to Questions

**Comments to the Authors:**

Reviewer #1: The manuscript describes an interesting and timely computational study of SARS-CoV-2 Spike protein variants and their impact on infectivity. The results will be very useful not only for the scientific community but also for the public health systems. The methodology presented could be of great help to predict the risk of new strains.

It’s well written. It would however beneficiate from a thorough review:

- Page 9: Explain the meaning of “May 08”

- In Figure 3, it is not clear which data are represented in the y-axis.

- Figure 8: Results showed in figure 8 should be better explained in the main text because there are some inconsistencies. Figure legend should be revised.

- It is not clear the usefulness of 3.5 section

- 3.6 section should be better included in Material & Method section

- A final conclusion should be included. Actual conclusions are more like a summary of the manuscript.

Reviewer #2: In the manuscript “Modelling conformational state dynamics and its role on infection for SARS-CoV-2 Spike protein variants” the authors utilize coarse-grained normal mode analyses to model the dynamics of Spike proteins and calculate transition probabilities between states for a number of Spike variants. The results predict an increase in open-state occupancy for the more infectious D614G via an increase in flexibility of the closed-state and decrease of flexibility of the open-state. The manuscript also presents a high throughput analysis of simulated single amino acid mutations on dynamic properties to seek potential hotspots and individual Spike variants that may be more infectious. The authors introduce a Markov model of occupancy of molecular states with transition probabilities derived from our analysis of dynamics that recapitulates experimental data on conformational state occupancies.

The biological problems addressed in this work are of clear fundamental and therapeutic interest and insights from computational approaches are certainly welcome to improve our understanding of the SARS-CoV-2 spike mechanisms and interactions.

Major points:

1.This is a fairly well-executed technical study describing an interesting combination of computational simulation tools to understand mechanisms of SARS-CoV-2 spike proteins in the native and mutant states. Although some of the presented results are certainly very interesting, the manuscript lacks organization, structure and a clearly formulated methodological objective. The overall presentation of the results is fragmented making difficult to understand the logic and methodological details of this work.

2. There is an enormous literature about this manuscript (both computational and experimental) that is only very briefly mentioned in Introduction. The authors should have critically assessed the previous studies and, more importantly, identify key issues and questions unanswered thus far.

3) The performed CG simulations do not apparently include the glycosylation of the spike, therefore strongly reducing the biological relevance of the entire work. Perhaps the authors should consider a model to mimic the glycosylated microenvironment in the framework of CG approaches. Although glycans are not supported by many CG methods which represents an important limitation in adopting this specific computational approach to study fusion viral proteins where glycosylation plays a key role, there exist CG methods ( such as Martini) where the energetic parameters have been recently extended to N-glycans. Glycans have been widely found to be crucial in the modulation of the spike conformational dynamics and should be considered for modeling of spike proteins. The inclusion of glycans to the model could potentially change the results and this comparison would be very important to substantiate the main findings and conclusions.

4) The authors claim that their results correctly model an increase in open-state occupancy for the more infectious D614G via an increase in flexibility of the closed-state and decrease of flexibility of the open-state.

There have been recently cryo-EM structures of a full-length G614 trimer (Structural impact on SARS-CoV-2 spike protein by D614G substitution. Science 2021, 372, 525-530) for distinct prefusion conformations in the closed, intermediate and 1-up open states (pdb ids 7krq, 7krs, 7krr) that characterized previously disordered regions in spike protein. This study supported the reduced shedding mechanism and suggested the increased stability of the G614 mutant. At the same time, another recent work in PNAS ( The effect of the D614G substitution on the structure of the spike glycoprotein of SARS-CoV-2. Proc. Natl. Acad. Sci. U. S. A. 2021, 118, e2022586118, pdb ids 7bnm, 7bnn, 7bno). These G614 mutant structures were more flexible and wide-open which is in line with the increased flexibility of the open state as proposed in the reviewed manuscript. These studies proposed different mechanisms, but it may reflect the diversity of conformational states adopted by the G614 mutant spike trimer.

Given simplicity of the elastic network models, I would suggest testing these structures ( or at least some of them) to try to reconcile conflicting mechanisms and also understand the effect of the G614 structures on the results and predictions.

5. Could the authors more clearly identify what makes their findings novel to biological community? What do the results of this study add to our current knowledge of the role of protein dynamics in these mechanisms?

6. It would be desirable to also use all-atom MD simulations for some of the studied systems to allow for a comparative analysis of protein flexibility. In general, the analysis of computational simulations are not sufficiently justified which weakens their connection with the biological evidence.

7. I believe that the authors should spend some time thinking how to strengthen the interface between experiment and computations in the manuscript to substantiate key findings.

8) Although the system is fascinating and computational approach is generally appropriate, the manuscript often reads as a set of disconnected observations rather than a cohesive story with the detailed analysis and insightful discussion.

9) The results often lack proper interpretation and integration with experiment to justify findings.

10). I believe that the authors should spend some time thinking how to strengthen the interface between experiment and computations in the manuscript to substantiate key findings.

Minor points:

1. The illustrations are often not sufficiently informative and generally very poor. Many of the plots and data cannot be seen at all. The authors should redesign and redo most of these figures and make them better organize, visible and informative with necessary annotations.

2. The manuscript is lacking a systematic statistical framework for assessing significance and quality of predictions. The authors should more clearly formulate and apply their statistical instruments along a common strategy to provide more confidence of quality and reproducibility of their results.

**Have the authors made all data and (if applicable) computational code underlying the findings in their manuscript fully available?**

Reviewer #2: Yes

PLOS authors have the option to publish the peer review history of their article (what does this mean?). If published, this will include your full peer review and any attached files.

Reviewer #1: No

Reviewer #2: No

**Have all data underlying the figures and results presented in the manuscript been provided?**

Reviewer #1: Yes
---

## [Editor Report · Decision Letter 1]

17 Jul 2021

Dear Dr. Najmanovich,

We are pleased to inform you that your manuscript 'Modelling conformational state dynamics and its role on infection for SARS-CoV-2 Spike protein variants' has been provisionally accepted for publication in PLOS Computational Biology.

I'm persuaded the new experimental data confirms your calculations and that your paper should be accepted.

Best regards,

Roland L. Dunbrack Jr., Ph.D.

Associate Editor

PLOS Computational Biology

Arne Elofsson

Deputy Editor

PLOS Computational Biology

---

## [Editor Report · Acceptance letter]

3 Aug 2021

PCOMPBIOL-D-21-00189R1 

Modelling conformational state dynamics and its role on infection for SARS-CoV-2 Spike protein variants

Dear Dr Najmanovich,

I am pleased to inform you that your manuscript has been formally accepted for publication in PLOS Computational Biology. Your manuscript is now with our production department and you will be notified of the publication date in due course.

With kind regards,

Katalin Szabo
